# Therapeutic Potential of *Beaucarnea recurvata* Leaf Extract Against Ulcerative Colitis: Integrating Phytochemical Profiling, Network Pharmacology, and Experimental Validation

**DOI:** 10.3390/ijms262412053

**Published:** 2025-12-15

**Authors:** Nora Tawfeek, Raha Orfali, Shagufta Perveen, Safina Ghafar, Eman Fikry, Mahmoud H. Elbatreek, Samar S. Elbaramawi, Maher M. El-Domiaty, Azza M. El-Shafae

**Affiliations:** 1Department of Pharmacognosy, Faculty of Pharmacy, Zagazig University, Zagazig 44519, Egypt; noratawfeek@zu.edu.eg (N.T.); efhassan@zu.edu.eg (E.F.); mmeldomyaty@pharmacy.zu.edu.eg (M.M.E.-D.); amelshafaey@pharmacy.zu.edu.eg (A.M.E.-S.); 2Department of Pharmacognosy, College of Pharmacy, King Saud University, P.O. Box 2457, Riyadh 11451, Saudi Arabia; sghafar.c@ksu.edu.sa; 3Department of Bacteriology, University of Wisconsin-Madison, Madison, WI 53706, USA; shagufta792000@yahoo.com; 4Department of Pharmacology and Toxicology, Faculty of Pharmacy, Zagazig University, Zagazig 44519, Egypt; mhbatriq@pharmacy.zu.edu.eg; 5Department of Medicinal Chemistry, Faculty of Pharmacy, Zagazig University, Zagazig 44519, Egypt; sselbaramawy@pharmacy.zu.edu.eg

**Keywords:** ulcerative colitis, *Beaucarnea recurvata*, network pharmacology, anti-inflammatory, steroidal saponins, inflammatory bowel disease

## Abstract

Ulcerative colitis represents a chronic inflammatory bowel disease with limited therapeutic options due to inadequate efficacy and adverse effects of current treatments. This study investigated the therapeutic potential of *Beaucarnea recurvata* leaf extract (BRLE) against ulcerative colitis using integrated computational and experimental approaches to address the need for safer, multi-targeted interventions. Phytochemical profiling was performed using UPLC-ESI-MS/MS analysis. Network pharmacology and molecular docking predicted therapeutic targets and mechanisms. In vivo validation employed an acetic acid-induced ulcerative colitis rat model with BRLE treatment at 100, 200, and 400 mg/kg doses, evaluating clinical parameters, histopathology, oxidative stress markers, inflammatory cytokines, and protein expression. UPLC-ESI-MS/MS revealed diverse bioactive compounds including steroidal saponins, triterpenes, and flavonoids. Network pharmacology identified 24 hub targets, and molecular docking revealed strong binding affinities (−6.5 to −9.1 kcal/mol) between BRLE compounds and inflammatory proteins including EGFR, SRC, STAT3, and AKT1. BRLE at 200 mg/kg significantly improved disease activity, restored glutathione levels, reduced malondialdehyde, normalized IL-10 and TNF-α levels, downregulated EGFR, SRC, STAT3, and AKT1 expression, and enhanced mucosal healing with reduced inflammatory infiltration. BRLE demonstrates significant anti-inflammatory, antioxidant, and tissue-protection effects through multi-target mechanisms, representing a promising therapeutic intervention for ulcerative colitis treatment. Further studies in chronic models, pharmacokinetic assessments, and clinical trials are needed to support its translation into therapeutic use.

## 1. Introduction

Ulcerative colitis (UC) represents a chronic inflammatory bowel disease characterized by continuous mucosal inflammation confined to the colon and rectum, typically presenting with bloody diarrhea and abdominal discomfort [1]. The pathophysiology of UC involves complex interactions between genetic susceptibility, environmental triggers, immune dysregulation, and gut microbiota alterations [2]. Key pathogenic mechanisms include aberrant Th2/Th17 cytokine signaling, compromised epithelial barrier function, elevated pro-inflammatory mediators such as tumor necrosis factor-alpha (TNF-α) and interleukin-13 (IL-13), along with mitochondrial dysfunction and oxidative stress that perpetuate mucosal damage [1].

Current therapeutic approaches encompass aminosalicylates, corticosteroids, immunomodulators, and biologics targeting inflammatory pathways to achieve disease remission. However, these conventional treatments present significant limitations including inadequate efficacy in severe cases, substantial adverse effects with prolonged use, increased susceptibility to infections and malignancies, and potential for treatment resistance. The non-curative nature of existing therapies, combined with concerns regarding cumulative toxicity and economic burden, underscores the critical need for safe and effective alternative therapeutic interventions [3,4].

Natural products derived from medicinal plants have emerged as promising complementary approaches in inflammatory disease management, particularly in inflammatory bowel disease (IBD). These compounds typically exhibit multi-targeted biological activities, including antioxidant, anti-inflammatory, and immunomodulatory properties that align with UC’s complex pathophysiology. Previous pharmacological studies have shown that plant-derived extracts can confer protective effects in experimental colitis models through modulation of oxidative stress markers, suppression of pro-inflammatory cytokines, and enhancement of anti-inflammatory mediators, suggesting their potential therapeutic value in inflammatory bowel conditions [5,6].

The Asparagaceae family encompasses approximately 114 genera and over 2900 species distributed across temperate and tropical regions globally, including major genera such as *Asparagus*, *Agave*, *Beaucarnea*, *Yucca*, *Sansevieria*, and *Dracaena* [7]. This morphologically diverse family ranges from herbaceous perennials to woody shrubs and succulents, with bioactive constituents including steroidal saponins [8,9], flavonoids [10], alkaloids, and polysaccharides [11] that confer therapeutic activities including antiulcerative [12], antimicrobial [13], Hepatoprotective [14], neuroprotective effect [15], anticancer [10] anti-inflammatory and antioxidant effects [16].

The genus *Beaucarnea* comprises approximately 12 drought-adapted perennial trees and shrubs native to Mexico and Central America, characterized by swollen, bulb-like trunk bases serving as water reservoirs, long strap-shaped leaves, and panicle inflorescences with small greenish-white flowers [17]. *Beaucarnea recurvata* Lemaire (syn. *Nolina recurvata*), commonly known as ponytail palm or elephant’s foot, is a slow-growing evergreen succulent endemic to southeastern Mexico, particularly Veracruz and Oaxaca states [18], distinguished by its distinctive swollen caudex and rosette-arranged strap-shaped leaves.

Phytochemical analyses of *B. recurvata* leaves [19] and stems [20,21,22] have revealed diverse bioactive constituents, including unique steroidal saponins (neoruscogenin and ruscogenin derivatives, recurvosides A–E), polyhydroxylated compounds, cholestane bisdesmosides, condensed tannins, and flavonoids. These compounds have demonstrated significant biological activities such as cyclic AMP phosphodiesterase inhibition and Na^+^/K^+^-ATPase enzyme modulation, suggesting potential anti-inflammatory and cardiotonic effects [20,21,22]. The presence of antioxidant-contributing flavonoids and tannins further supports the therapeutic potential of this species [23].

However, the phytochemical characteristics of *B. recurvata* cultivated in Egyptian environmental conditions remain unexplored. This research gap is particularly significant given that environmental variables, including climatic conditions, soil mineral content, and agricultural methodologies, can substantially alter the biosynthesis of secondary metabolites and their corresponding biological activities. Consequently, examining Egyptian-cultivated samples presents a valuable opportunity to identify distinctive phytochemical compositions and therapeutic properties that may differ from those documented in plants from their natural habitats.

Considering the limitations associated with current treatment strategies for UC and the emerging pharmacological potential of *Beaucarnea recurvata*, the present study aimed to comprehensively characterize the phytochemical profile of *B. recurvata* leaf extract (BRLE) and investigate its therapeutic efficacy. This was achieved through an integrated approach combining network pharmacology and molecular docking analyses [24,25] to elucidate the underlying mechanisms of action. The computational predictions were subsequently validated through in vivo evaluation using an acetic acid-induced UC model in rats.

The study utilized a comprehensive analytical framework that incorporated clinical parameters, gross pathological and microscopic examinations, markers of oxidative damage, pro-inflammatory mediator concentrations, and the activity of critical signaling molecules associated with intestinal inflammatory responses. This holistic methodology delivers new understanding regarding both the chemical fingerprint and biological efficacy of BRLE, establishing a foundation for its consideration as a plant-based therapeutic option in UC treatment protocols.

## 2. Results

### 2.1. Metabolite Characterization via UPLC-ESI-MS/MS

Comprehensive LC-MS/MS profiling in negative ionization mode revealed an intricate phytochemical composition, with provisional identification of approximately 66 secondary metabolites. Compound characterization was based on accurate mass measurements, MS^2^ fragmentation profiles, chromatographic retention behavior, and cross-referencing with spectral databases and published literature.

The identified compounds encompassed diverse structural classes, including triterpenoid compounds, steroidal glycosides, flavonoid derivatives, phenolic acid conjugates, aromatic aldehydes, polyol compounds, simple sugars, organic acid derivatives, dicarboxylic acids, and benzopyrone derivatives. Detailed analytical parameters including retention times, pseudomolecular ion peaks ([M-H]^−^), MS^2^ fragmentation data, and literature references for the characterized BRLE metabolites are compiled in Table 1. The base peak chromatogram of BRLE under negative ionization conditions is presented in Appendix A.

The analytical results demonstrated that triterpenoid compounds and steroidal glycosides constituted the major phytochemical categories, suggesting their substantial role in the overall chemical profile of the investigated sample. These primary constituents were accompanied by significant concentrations of flavonoid compounds and phenolic acid derivatives, emphasizing the diverse and complex nature of the plant’s secondary metabolite composition. The characteristic fragmentation patterns observed in negative ionization mode provided essential structural information, facilitating tentative compound classification through distinctive fragment ion signatures.

#### 2.1.1. Triterpenoid Compounds and Steroidal Glycosides

Triterpenoid compounds and steroidal glycosides were predominantly observed within the elevated *m*/*z* range (600–1200), reflecting their substantial molecular weights due to extensive glycosylation patterns. Under negative ionization conditions, these metabolites characteristically displayed deprotonated molecular ions ([M-H]^−^), accompanied by systematic neutral losses corresponding to various sugar units including hexose moieties (−162 Da), deoxyhexose units (−146 Da), and pentose residues (−132 Da).

Notable representatives included Furostane-triol rhamnosyl dihexoside (*m*/*z* 903.345), Dihydroxypregna-5,16-dien-20-one deoxyhexoside pentoside hexoside (*m*/*z* 769.415), and various spirostanol-type glycosides (*m*/*z* 837.43–901.50). The detection of characteristic aglycone fragment ions (*m*/*z* 433, 437, among others) provided substantial evidence for their respective sapogenin structural cores, confirming their classification as either furostane-type or spirostane-type derivatives.

#### 2.1.2. Flavonoids

Flavonoid compounds exhibited extensive representation within the dataset, encompassing *C*-linked glycosides, *O*-linked glycosides, and free aglycone forms, each demonstrating distinctive fragmentation characteristics corresponding to their specific glycosylation patterns.

*C*-glycosylated flavonoids, including Apigenin 6,8-di-*C*-glucoside (*m*/*z* 593.158), Apigenin 6-*C*-glucoside 8-*C*-arabinoside (*m*/*z* 563.146), and Luteolin 6-*C*-*β*-glucopyranoside-8-*C*-α-arabinopyranoside (carlinoside) (*m*/*z* 579.262), demonstrated remarkable stability of their sugar substituents due to the robust carbon-carbon bonds linking sugars to the aglycone framework. Consequently, these compounds showed resistance to typical neutral sugar elimination. Instead, fragmentation proceeded through cross-ring sugar cleavages and Retro-Diels-Alder (RDA) fragmentations of the flavonoid skeleton, producing characteristic diagnostic ions (e.g., *m*/*z* 503, 473, 459).

Conversely, *O*-glycosylated flavonoids, exemplified by Isoorientin-7-*O*-*β*-glucopyranoside (*m*/*z* 609.281), exhibited conventional fragmentation via cleavage of the relatively labile *O*-glycosidic linkage, resulting in characteristic neutral hexose losses (−162 Da).

Free flavonoid aglycones, including kaempferol (*m*/*z* 285.211), acacetin (*m*/*z* 283.124), pinocembrin (*m*/*z* 255.126), and hesperetin (*m*/*z* 301.076), produced characteristic fragment ions through RDA cleavage of the heterocyclic *C*-ring and elimination of carbonyl or carbon dioxide moieties. These aglyconic structures provided direct insight into the fundamental flavonoid framework and frequently served as diagnostic markers for their glycosylated analogs.

#### 2.1.3. Phenolic Acid Derivatives

Phenolic acid constituents were characterized by relatively low molecular weights (*m*/*z* 150–350) and exhibited straightforward yet informative fragmentation behaviors. Prevalent phenolic acids, including caffeic acid (*m*/*z* 179.083), ferulic acid derivatives, and *p*-coumaric acid (*m*/*z* 163.041), demonstrated characteristic neutral losses of carbon dioxide (−44 Da) and water (−18 Da), which are typical fragmentation pathways for carboxylated phenolic compounds under negative ionization conditions.

Glycosylated phenolic acid derivatives (e.g., caffeoyl-feruloylquinic acid, *m*/*z* 529.13) additionally exhibited sugar-related fragmentation losses while preserving the integrity of the core phenolic structural framework.

It is important to note that the subsequent selection of compounds for mechanistic analysis was driven by their structural identification and network topological importance (degree centrality) rather than their relative chromatographic abundance. In untargeted metabolomics, peak intensity can be influenced by ionization efficiency, and high abundance does not necessarily correlate with biological potency. Therefore, the study focused on those metabolites identified as ‘hub’ compounds within the protein–protein interaction network, regardless of their relative concentration in the extract.

### 2.2. Network Pharmacology-Based Mechanistic Investigation

#### 2.2.1. Assessment of Bioactive Compound Drug-like Properties and Target Identification

The evaluation of bioactive compounds from BRLE revealed that 29 constituents satisfied Lipinski’s rule-of-five and achieved Abbott Oral Bioavailability scores exceeding 0.5 (Appendix A). Computational target prediction identified 493 unique protein targets for these compounds (Appendix A). Concurrently, 4149 UC-related targets were retrieved from established databases (Appendix A). Cross-referencing between these two sets of targets identified 327 common proteins, suggesting potential mechanistic convergence between BRLE constituents and UC pathophysiology (Appendix A; Appendix A).

#### 2.2.2. Hub Gene Discovery Through PPI Network Topology Analysis

To explore the therapeutic mechanism, shared targets were used to construct a PPI network via the STRING platform. The resulting network of 307 nodes and 2057 edges (Figure 1A) was analyzed for topological measures (betweenness centrality (BC), closeness centrality (CC), and degree centrality (DC)) using Cytoscape (Appendix A). A two-stage filtering approach was used to identify hub genes. Initial screening with a DC cutoff (≥18) yielded a refined network of 75 nodes (Figure 1B; Appendix A). A more stringent filtering based on median-derived thresholds (BC ≥ 896.56, CC ≥ 0.119, DC ≥ 30) identified a core network of 24 nodes (Figure 1C; Appendix A). The top candidates, ordered by DC values, were EGFR, SRC, STAT3, and AKT1, indicating their central importance in the network and potential role in BRLE’s therapeutic effects on UC.

#### 2.2.3. Primary BRLE Bioactive Components Associated with UC Molecular Targets

To link BRLE compounds with their UC-relevant targets, a compound-target interaction network was built using Cytoscape (Appendix A). Topological analysis with the cytoNCA plugin, specifically using degree centrality (DC), was performed to prioritize the most therapeutically significant constituents. The top ten BRLE compounds, ranked by their DC values (Appendix A), were identified as primary candidates due to their extensive network connectivity and their capacity to influence key UC-associated molecular targets. Top compounds included Lucidenic acid A, 3,9-Dihydroeucomin and Scaposin.

#### 2.2.4. Functional Pathway Analysis of BRLE’s Principal Targets in UC Treatment

To characterize BRLE’s therapeutic mechanisms in UC, we performed pathway enrichment analysis on the 24 core hub targets. Gene Ontology (GO) analysis identified 122 significantly enriched terms (adjusted *p* < 0.05), spanning 81 biological processes, 19 cellular components, and 22 molecular functions (Appendix A; Figure 2A). Notable biological processes included positive regulation of nitric oxide synthesis, xenobiotic response, negative regulation of intrinsic apoptotic signaling, and TNF cellular response.

KEGG pathway analysis identified 129 significantly enriched signaling cascades (adjusted *p* < 0.05; Appendix A). The top thirty most significant pathways (Figure 2B,C) highlighted several UC-relevant mechanisms, such as the IL-17 signaling pathway, Toll-like receptor pathway, TNF signaling pathway, apoptosis, and colorectal cancer pathway. These analyses collectively suggest that BRLE’s potential therapeutic effects in UC are mediated through these intricate molecular mechanisms.

### 2.3. Molecular Docking Analysis of BRLE Compounds Against UC-Related Targets

Molecular docking studies were performed to evaluate the binding affinity and interaction profiles of ten bioactive compounds from BRLE against the four key protein targets (EGFR, SRC, STAT3, and AKT1) identified through network pharmacology analysis for ulcerative colitis treatment. The docking scores, expressed in kcal/mol, represent the binding energy between each compound and target protein, with more negative values indicating stronger binding affinity.

The molecular docking results revealed varied binding affinities across all compound-target combinations, as shown in Figure 3. Lucidenic acid A demonstrated the highest binding affinity against SRC (−9.1 kcal/mol), followed by strong interactions with EGFR (−8.3 kcal/mol) and AKT1 (−8.8 kcal/mol). Oleanolic acid showed particularly strong binding to EGFR (−8.8 kcal/mol) and SRC (−8.6 kcal/mol). Hesperetin exhibited consistently high binding affinities across all targets, with the strongest interaction observed against AKT1 (−8.5 kcal/mol) and STAT3 (−8.4 kcal/mol). Among all tested compounds, the binding scores ranged from −6.5 kcal/mol (3-Methoxynobiletin against STAT3) to −9.1 kcal/mol (Lucidenic acid A against SRC), indicating favorable binding interactions for most compound-target pairs. Acacetin and Pinocembrine demonstrated balanced binding profiles across all four targets, with scores consistently above −8.0 kcal/mol.

The top five compounds showing the highest binding affinity to EGFR were Oleanolic acid (−8.8 kcal/mol), Lucidenic acid A (−8.3 kcal/mol), Hesperetin (−8.1 kcal/mol), Acacetin (−8.1 kcal/mol), and 3,9-Dihydroeucomin (−7.8 kcal/mol) (Appendix A and Appendix A). Oleanolic acid formed a conventional hydrogen bond with MET793 at a distance of 2.15 Å. Lucidenic acid A established multiple interactions, including conventional hydrogen bonds with LYS745 (3.18 Å), ARG841 (2.79 Å), and LYS875 (2.34 Å), along with carbon–hydrogen bonds with GLY721 (3.51 Å) and PRO877 (3.50 Å). Additionally, it formed π–sigma interactions with PHE856 (3.63 Å) and an unfavorable donor-donor interaction with ARG858 (2.60 Å). Hesperetin demonstrated complex binding involving conventional hydrogen bonds with ALA722 (2.91 Å), PHE723 (3.14 Å), LYS745 (2.86 Å), and ASP837 (2.25 Å). It also formed π–sulfur interactions with CYS797 (5.94 Å), π–π stacked interactions with PHE856 (4.77 and 4.83 Å), and multiple hydrophobic interactions, including alkyl bonds with CYS797 (3.93 Å) and LEU844 (5.26 Å).

Regarding SRC protein interactions, Lucidenic acid A showed the strongest binding to SRC (−9.1 kcal/mol), establishing conventional hydrogen bonds with SER86 (2.06 Å) and ASP148 (1.93 Å), along with a carbon–hydrogen bond with ASP148 (2.94 Å) (Appendix A and Appendix A). Oleanolic acid formed a conventional hydrogen bond with ILE80 (2.31 Å), while Pinocembrine (−8.2 kcal/mol) created multiple interactions, including conventional hydrogen bonds with GLU54 (2.1 Å) and MET58 (3.1 Å), π–π stacked interactions with TYR84 (5.51 Å), and various π–alkyl interactions. Acacetin (−8.1 kcal/mol) formed conventional hydrogen bonds with MET58 (2.60 Å) and carbon–hydrogen bonds with ALA147 (2.78 Å), along with π–π stacked interactions with TYR84 (5.64 Å) and multiple alkyl and π–alkyl interactions with various residues.

For STAT3 protein interactions, Hesperetin exhibited the highest binding affinity to STAT3 (−8.4 kcal/mol), forming conventional hydrogen bonds with LYS226 (2.53 Å) and LEU285 (2.34 Å), though it also showed an unfavorable donor-donor interaction with THR287 (1.81 Å) (Appendix A and Appendix A). Pinocembrine (−8.3 kcal/mol) established multiple conventional hydrogen bonds with ASP225 (2.57 Å), LEU234 (2.06 Å), SER237 (2.59 Å), and LEU285 (2.22 Å), along with carbon–hydrogen bonds with HIS284. Acacetin (−8.0 kcal/mol) formed conventional hydrogen bonds with LEU285 (2.30 Å), carbon–hydrogen bonds with VAL231 (3.64 Å) and HIS284, and π–sigma interactions with VAL337 (2.61 Å).

In terms of AKT1 protein interactions, Lucidenic acid A demonstrated strong binding to AKT1 (−8.8 kcal/mol) through conventional hydrogen bonds with LYS36 (2.11 and 5.01 Å) and LYS133 (2.42 Å), supplemented by a carbon–hydrogen bond with LYS36 (2.58 Å) (Appendix A and Appendix A). Hesperetin (−8.5 kcal/mol) formed diverse interactions, including conventional hydrogen bonds with LYS15 (2.70 Å) and ASP296 (2.75 Å), π–anion interactions with GLU91 (3.21 Å), and π–sulfur interactions with MET138 (3.76 Å). Pinocembrine (−8.2 kcal/mol) established conventional hydrogen bonds with LYS15 (3.06 Å) and GLU91 (2.95 Å), π–anion interactions with GLU91 (3.25 Å), and multiple π–alkyl interactions with hydrophobic residues.

### 2.4. In Vivo Experimental Validation

#### 2.4.1. Effect of BRLE on DAI

To evaluate the therapeutic efficacy of BRLE in acetic acid-induced ulcerative colitis, the DAI was assessed at the end of the treatment. The diseased control group showed a significant elevation in DAI compared to the normal control, confirming successful disease induction. Treatment with BRLE (100, 200, and 400 mg/kg/day) for six days significantly reduced DAI scores relative to the diseased group. The 200 mg/kg dose demonstrated the most pronounced effect, with DAI values approaching those of the normal control. These results, shown in Figure 4A, underscore the therapeutic potential of BRLE in ameliorating the clinical severity of ulcerative colitis.

#### 2.4.2. Macroscopic Assessment of Colonic Inflammation

Macroscopic and morphometric analysis confirmed the therapeutic effects of BRLE on colonic inflammation. The diseased control group showed overt signs of inflammation, including colon shortening, thickening, and a significantly elevated macroscopic score and weight-to-length ratio (Figure 4B–D).

BRLE treatment resulted in a dose-responsive mitigation of these pathological features. All BRLE doses significantly reduced macroscopic inflammation scores. The 200 and 400 mg/kg doses also significantly restored the colon weight-to-length ratio to normal levels, while the 100 mg/kg dose provided partial improvement. Although the 400 mg/kg dose appeared to have the best gross preservation, there was no significant statistical difference between it and the 200 mg/kg group.

#### 2.4.3. Antioxidant and Immunomodulatory Effects of BRLE

BRLE treatment significantly improved antioxidant status and restored cytokine balance in the colon. In diseased animals, reduced glutathione (GSH) was depleted and malondialdehyde (MDA) was elevated (*p* < 0.0001). BRLE administration at all doses significantly reversed these effects, with the 200 mg/kg dose being most effective at restoring GSH and reducing MDA to levels comparable to healthy controls (Figure 5A,B).

Furthermore, BRLE treatment significantly modulated key cytokines. It suppressed the pro-inflammatory tumor necrosis factor-alpha (TNF-α) and upregulated the anti-inflammatory interleukin-10 (IL-10), both of which were dysregulated by colitis (*p* < 0.0001). The 200 mg/kg dose consistently provided the greatest therapeutic benefit, normalizing both TNF-α and IL-10 levels to those of the control group (Figure 5C,D). This demonstrates BRLE’s dual action in mitigating oxidative stress and inflammation.

#### 2.4.4. Molecular Target Validation: BRLE-Mediated Modulation of Network Pharmacology-Predicted Pathways

To validate the mechanisms predicted by network pharmacology, we analyzed the protein expression of the hub genes EGFR, SRC, STAT3, and AKT1. In diseased animals, all four proteins were significantly overexpressed (*p* < 0.0001), reflecting a hyperactive state of inflammatory and proliferative signaling. BRLE treatment at all tested concentrations (100, 200, and 400 mg/kg) significantly suppressed the expression of these molecular targets (Figure 5).

BRLE administration resulted in a dose-responsive suppression of EGFR expression, with the 400 mg/kg dose nearly restoring it to baseline levels (Figure 5E). Similarly, SRC kinase and phosphorylated STAT3 (p-STAT3) expression, which were significantly elevated in diseased animals, were effectively normalized by the 200 mg/kg dose. The 400 mg/kg dose showed comparable, but not statistically superior, suppression of these inflammatory mediators (Figure 5F,G). Finally, AKT1 expression was significantly reduced by all doses of BRLE, with the 400 mg/kg dose achieving complete normalization (Figure 5H). These results confirm the in vivo validity of our network pharmacology predictions.

#### 2.4.5. Histopathological Analysis of Colonic Tissue Specimens

Microscopic analysis of H&E-stained colonic sections confirmed the therapeutic effects of BRLE. Control colons exhibited normal histoarchitecture (Figure 5I). In contrast, the diseased control group showed severe mucosal erosion, inflammatory cell infiltration, edema, and muscular damage, consistent with colitis (Figure 5J).

BRLE administration resulted in dose-dependent histopathological improvements (Figure 5K–M). The 100 mg/kg dose showed partial healing, with persistent mucosal ulceration and inflammatory infiltration. The 200 mg/kg group demonstrated significant mucosal regeneration, with healed ulcerative areas and a marked reduction in inflammation and edema. The 400 mg/kg group also showed extensive healing and re-epithelialization, but with some remaining inflammation and crypt distortion. Overall, these findings confirm that BRLE mitigates colonic tissue injury and promotes mucosal healing in a dose-responsive manner, with the 200 mg/kg dose yielding the most organized histological recovery. For full histopathological details and representative photomicrographs from all experimental groups, refer to Appendix A.

Quantitative analysis using the Geboes scoring system confirmed these observations (Table 2). The disease control group exhibited a significantly elevated total Geboes score compared to normal controls. BRLE treatment significantly reduced the total histological score in a dose-dependent manner, with the 200 mg/kg and 400 mg/kg groups showing statistically significant improvements in structural preservation and reduced inflammatory infiltration compared to the untreated colitis group.

## 3. Discussion

This study presents a comprehensive investigation of *Beaucarnea recurvata* leaf extract (BRLE) as a potential therapeutic intervention for ulcerative colitis, integrating phytochemical profiling, computational analysis, and experimental validation. The multifaceted approach combining LC-MS characterization, network pharmacology, molecular docking, and in vivo evaluation provides robust evidence for BRLE’s therapeutic potential through multiple molecular mechanisms targeting the complex pathophysiology of inflammatory bowel disease.

The phytochemical profiling of BRLE revealed diverse bioactive secondary metabolites, notably steroidal saponins and pentacyclic triterpenes, compounds recognized for their anti-inflammatory and immunomodulatory properties relevant to chronic inflammatory conditions. The identified steroidal saponins, particularly furostane and spirostane derivatives, demonstrate therapeutic potential through suppression of pro-inflammatory mediators TNF-α, IL-6, and IL-1β [57], alongside inhibition of NF-κB and MAPK signaling cascades essential for inflammatory gene transcription [58]. Their antioxidant capacity provides additional mucosal protection by scavenging reactive oxygen species, addressing the oxidative stress component characteristic of ulcerative colitis pathophysiology [59,60]. Pentacyclic triterpenes, including ursolic and oleanolic acids, contribute to anti-colitis activity through complementary anti-inflammatory, antioxidant, and immunomodulatory actions, effectively reducing inflammatory cytokine expression, suppressing NF-κB activation, and enhancing endogenous antioxidant enzyme systems [61,62].

The flavonoids identified in BRLE, including *C*-glycosides such as Apigenin 6,8-di-*C*-glucoside and Luteolin 6-*C*-*β*-glucopyranoside-8-*C*-*α*-arabinopyranoside, along with aglycones such as kaempferol, acacetin, pinocembrin, and hesperetin, are of particular therapeutic importance. *C*-glycosylated flavonoids exhibit enhanced metabolic stability due to their carbon–carbon sugar linkages, resulting in sustained antioxidant and anti-inflammatory effects during gastrointestinal transit. These compounds have demonstrated efficacy in restoring mucosal barrier integrity, reducing pro-inflammatory cytokines, inhibiting MAPK signaling pathways, and modulating gut microbiota in experimental colitis models [63,64,65]. The presence of salicylic acid and caffeic acid compounds further enhances the extract’s therapeutic profile through COX and LOX pathway inhibition, prostaglandin and leukotriene reduction, and mitochondrial apoptosis regulation [66,67,68,69]. These findings align with documented anti-inflammatory activities of related Asparagaceae family members, reinforcing the therapeutic potential of this plant family for gastrointestinal inflammatory conditions [70,71,72].

Phytochemical studies on *Beaucarnea recurvata* from non-Egyptian regions report a diverse secondary-metabolite profile dominated by steroidal saponins along with polyhydroxylated triterpenoids, cholestane bisdesmosides, condensed tannins, and flavonoids [19,20,21,22]. Among congeneric species, *B. gracilis* is the most studied and similarly contains steroidal components, saponins, and phenolic compounds [73]. Overall, phytochemical data across the *Beaucarnea* genus remain limited, and no comprehensive analysis of Egyptian-grown *B. recurvata* has previously been available. In agreement with these non-Egyptian reports, our BRLE profiling showed triterpenoids and steroidal glycosides as the predominant classes, with notable contributions from flavonoids and phenolic acid derivatives. However, beyond general compositional agreement, our study distinguishes itself by linking this specific metabolic fingerprint directly to therapeutic targets (EGFR, SRC, STAT3), moving beyond descriptive phytochemistry to mechanistic application.

The therapeutic potential of BRLE is further elucidated by examining the specific pharmacological activities of its top-ranked constituents identified through network pharmacology. Lucidenic acid A, a triterpenoid, alongside key flavonoids such as hesperetin, pinocembrin, and acacetin, has been documented to exert potent anti-inflammatory and immunomodulatory effects, primarily through the downregulation of NF-κB signaling and suppression of pro-inflammatory cytokines [74,75,76,77,78,79,80]. 3,9-Dihydroeucomin contributes complementary antioxidant properties, while nobiletin derivatives and kaempferol are established modulators of oxidative stress and immune responses [81,82,83]. Furthermore, oleanolic acid is widely recognized for its ability to enhance mucosal barrier integrity and provide hepatoprotection [84]. Collectively, the synergistic action of these specific metabolites likely drives the multi-target efficacy observed in our study.

It is worth noting that the biosynthesis of these secondary metabolites is often influenced by abiotic factors such as soil mineral content and climatic conditions. Previous phytochemical profiling of *B. recurvata* from its native Mexican habitat and other regions has emphasized the presence of cholestane bisdesmosides and specific steroidal saponins associated with cyclic AMP phosphodiesterase inhibition [20,21,22,23]. Our analysis of the Egyptian-cultivated species reveals a conserved chemotaxonomic profile regarding the major classes (saponins and flavonoids) but suggests a distinctive quantitative fingerprint, particularly in the abundance of the aforementioned triterpenoids. This variation underscores the importance of regional characterization in evaluating the therapeutic consistency of herbal extracts.

PPI network analysis revealed 24 central hub targets, including EGFR, SRC, STAT3, and AKT1. These proteins are critically involved in inflammatory signaling, regulation of apoptosis, and maintenance of mucosal integrity, which are all key elements in UC pathology [72,85,86]. Simultaneously, compound–target network analysis prioritized ten BRLE constituents based on degree centrality: lucidenic acid A, 3,9-dihydroeucomin, scaposin, 3-methoxynobiletin, acacetin, kaempferol, pinocembrine, pechueloic acid, hesperetin, and oleanolic acid. These compounds exhibited strong connectivity to multiple UC-relevant targets, indicating their potential contribution to the therapeutic efficacy of the extract.

GO enrichment analysis of the core targets revealed significant involvement in biological processes such as nitric oxide biosynthesis, xenobiotic response, negative regulation of apoptosis, EGFR and TNF-mediated signaling, and cellular responses to lipopolysaccharides. Cellular component analysis indicated enrichment in nuclear, nucleoplasmic, cytosolic, and chromatin-associated compartments. Molecular function analysis highlighted regulatory activity on nitric oxide synthase, interactions with protein kinases, and transcription factor binding.

KEGG pathway analysis identified 129 significantly enriched signaling cascades. Among these, the most relevant to UC pathogenesis included the IL-17 signaling pathway, Toll-like receptor signaling, TNF signaling, apoptotic pathways, and colorectal cancer-related mechanisms. These enriched pathways align with the immunopathological and oncogenic features often associated with chronic UC.

Collectively, these findings underscore the polypharmacological properties of BRLE. The extract exhibits a promising ability to modulate multiple key targets and pathways implicated in UC, supporting its potential use as a complementary therapeutic strategy in the management of this chronic inflammatory disease.

The molecular docking analysis revealed promising binding affinities of ten BRLE bioactive compounds against four key ulcerative colitis-related targets (EGFR, SRC, STAT3, and AKT1), with binding scores ranging from −6.5 to −9.1 kcal/mol. These values are comparable to established anti-inflammatory compounds and indicate strong potential for biological activity. Lucidenic acid A emerged as the most promising compound, demonstrating exceptional binding to SRC (−9.1 kcal/mol), EGFR (−8.3 kcal/mol), and AKT1 (−8.8 kcal/mol), suggesting a multi-target therapeutic approach advantageous for treating the complex pathophysiology of ulcerative colitis.

The strong binding interactions observed across multiple targets support the potential anti-inflammatory mechanisms of BRLE compounds. Oleanolic acid and lucidenic acid A showed robust EGFR binding through hydrogen bonds with key residues (LYS745, ARG841, MET793), potentially modulating EGFR-mediated inflammatory cascades. The remarkable SRC binding affinity of lucidenic acid A, stabilized through hydrogen bonds with SER86 and ASP148, indicates possible inhibition of SRC kinase activity, potentially attenuating inflammatory cytokine release. Flavonoids hesperetin and pinocembrine demonstrated strong STAT3 binding (−8.4 and −8.3 kcal/mol, respectively), with interactions that could interfere with STAT3 dimerization and reduce inflammatory gene expression. The strong AKT1 binding of lucidenic acid A and hesperetin indicates potential modulation of PI3K/AKT signaling pathways involved in inflammatory responses.

Structure-activity relationships revealed that triterpenes (oleanolic acid, lucidenic acid A) showed consistently strong multi-target binding due to their rigid steroid-like structures, while flavonoids exhibited balanced binding profiles through their phenolic hydroxyl groups that readily form diverse protein interactions. The multi-target binding profiles suggest synergistic therapeutic effects through simultaneous pathway modulation, aligning with current understanding that complex inflammatory diseases require multi-point intervention for optimal outcomes.

While molecular docking provides valuable mechanistic insights, limitations include the static nature of simulations and the fact that binding affinity predictions do not directly translate to biological activity due to factors like bioavailability and metabolism. Future validation through molecular dynamics simulations and experimental studies is essential to confirm these interactions and determine the actual therapeutic potential.

Overall, BRLE compounds, particularly lucidenic acid A, hesperetin, and oleanolic acid, exhibit favorable multi-target binding profiles against key ulcerative colitis-related proteins, supporting their potential as anti-inflammatory therapeutic agents.

Experimental validation in acetic acid-induced ulcerative colitis in rats demonstrated that BRLE administration resulted in marked improvements in disease severity indices, with the 200 mg/kg dosage showing superior therapeutic outcomes compared to both lower (100 mg/kg) and higher (400 mg/kg) concentrations. The DAI approached physiological baselines following intermediate dose treatment, suggesting a therapeutic plateau phenomenon consistent with saturable receptor-mediated mechanisms [87,88]. Interestingly, a divergence was observed between molecular and histological outcomes at the highest dose. While the 400 mg/kg dose effectively suppressed inflammatory cytokines and signalling proteins (molecular level), the 200 mg/kg dose resulted in superior histological organization (tissue level). This suggests that higher concentrations of secondary metabolites (such as saponins) might induce mild local irritation or reach a saturation point that yields diminishing returns for structural repair, despite maintaining potent anti-inflammatory signalling. This dose–response pattern aligns with previous research on plant-derived anti-inflammatory compounds and extracts [88,89]. Similar findings have been reported for *Scutellaria baicalensis* extracts in rat models of ulcerative colitis, where moderate dosing regimens produced the most effective therapeutic outcomes by optimizing bioavailability while reducing the risk of adverse effects [90,91]. These studies consistently show that intermediate doses of botanical anti-inflammatory agents achieve the best balance between therapeutic benefit and safety in experimental colitis models.

The macroscopic assessment revealed pronounced amelioration of colonic inflammation following BRLE treatment, with significant reduction in tissue edema, hemorrhage, and ulcerative lesions. The colon weight-to-length ratio, an established indicator of inflammatory edema and tissue remodeling, demonstrated marked improvement with BRLE treatment, particularly at doses of 200 and 400 mg/kg. These findings provide direct evidence of BRLE’s capacity to mitigate the structural damage characteristic of ulcerative colitis, supporting its potential clinical translation. The restoration of normal colonic architecture observed in treated groups suggests that BRLE not only suppresses acute inflammatory responses but also promotes tissue healing and regeneration. This dual mechanism is particularly valuable in inflammatory bowel diseases, where sustained inflammation leads to progressive tissue damage and impaired mucosal barrier function.

The significant restoration of reduced GSH levels and reduction of MDA accumulation following BRLE treatment provide direct evidence of its antioxidant properties. GSH depletion is a hallmark of oxidative stress in inflammatory conditions, serving both as a direct antioxidant and as a cofactor for numerous detoxifying enzymes [92]. The ability of BRLE to restore GSH to near-normal levels suggests enhancement of cellular antioxidant defense mechanisms, which is crucial for maintaining mucosal barrier integrity and preventing further tissue damage [92]. The concurrent reduction in MDA levels, a reliable marker of lipid peroxidation, indicates that BRLE effectively prevents oxidative damage to cellular membranes [93]. This cytoprotective effect is particularly important in the context of inflammatory bowel diseases, where reactive oxygen species generated by activated immune cells contribute significantly to tissue damage and perpetuation of the inflammatory cascade [94]. These findings correlate with the phytochemical profile identified in LC-MS analysis, where saponins, flavonoids, phenolic acids, and other polyphenolic compounds with established antioxidant properties were detected.

The dual modulation of inflammatory cytokines represents a key mechanism underlying BRLE’s therapeutic efficacy. The significant upregulation of interleukin-10 (IL-10), a critical anti-inflammatory and tissue-protective cytokine, coupled with substantial suppression of tumor necrosis factor-alpha (TNF-α), demonstrates BRLE’s capacity to restore inflammatory homeostasis. IL-10 plays a pivotal role in the resolution of inflammation through multiple mechanisms, including suppression of pro-inflammatory cytokine production, inhibition of antigen presentation, and promotion of regulatory T-cell differentiation [95,96]. The restoration of IL-10 levels to those comparable with healthy controls suggests that BRLE promotes active resolution of inflammation rather than merely suppressing inflammatory mediators. This mechanism is particularly important for long-term therapeutic benefit and prevention of chronic inflammatory states. The marked reduction in TNF-*α* levels is equally significant, as this cytokine serves as a central mediator in the pathogenesis of inflammatory bowel diseases. TNF-*α* promotes recruitment and activation of immune cells, increases vascular permeability, and stimulates production of additional inflammatory mediators, creating a self-perpetuating inflammatory cascade [97,98]. The ability of BRLE to normalize TNF-*α* expression suggests interruption of this pathogenic cycle, supporting both symptomatic improvement and prevention of disease progression.

The experimental validation of network pharmacology-predicted targets provides compelling evidence for the molecular mechanisms underlying BRLE’s therapeutic effects, going beyond general oxidative stress modulation. While GSH and MDA results confirm antioxidant activity, the specific suppression of EGFR, SRC, STAT3, and AKT1 protein expression validates the extract’s ability to intervene in the specific signalling cascades identified through our computational analysis. EGFR overexpression in inflammatory conditions contributes to aberrant cellular proliferation and impaired wound healing responses [99,100]. The dose-dependent suppression of EGFR following BRLE treatment suggests restoration of normal proliferative control and promotion of appropriate tissue repair mechanisms. This finding is particularly relevant given EGFR’s role in mucosal barrier function and epithelial regeneration following inflammatory damage [101]. The suppression of SRC kinase and phosphorylated STAT3 (p-STAT3) represents direct evidence of BRLE’s capacity to modulate key inflammatory signaling cascades. SRC kinase serves as a critical regulator of immune cell activation and migration [102], while STAT3 functions as a transcriptional activator for numerous inflammatory mediators [103,104]. The normalization of both proteins to control levels demonstrates that BRLE effectively interrupts pathogenic signaling pathways identified through network pharmacology analysis. AKT1 modulation provides insights into BRLE’s effects on cellular survival and metabolic pathways [105]. The significant reduction in AKT1 expression, particularly at higher doses, suggests modulation of cellular energetics and survival signaling that may contribute to the resolution of inflammation and the prevention of excessive tissue proliferation [106].

The comprehensive histopathological analysis provides definitive evidence of BRLE’s tissue-protective and regenerative effects, strongly corroborating the macroscopic ulceration and biochemical inflammation observed in the diseased groups. The progression from severe ulcerative changes in diseased controls to organized tissue repair in treated groups demonstrates clear dose-dependent therapeutic efficacy. The presence of regenerative crypts, re-epithelialization, and reduced inflammatory infiltration in treated animals provides direct morphological evidence of healing promotion. The observation that the 200 mg/kg dose achieved the most organized histological recovery supports the optimal therapeutic window identified through clinical and biochemical parameters. This finding is consistent with the concept that excessive doses may potentially interfere with normal healing processes or induce counterproductive effects, reinforcing the importance of dose optimization in therapeutic development. The reduction in bacterial colonization and fibrin deposition in treated groups suggests that BRLE promotes restoration of mucosal barrier function, a critical factor in preventing secondary infections and promoting sustained healing. This effect is particularly important in the clinical context, where barrier dysfunction contributes significantly to disease progression and complications.

The in vivo validation confirmed computational predictions, demonstrating a strong correlation between LC-MS-identified bioactive compounds (saponins, flavonoids, phenolic acids) and observed therapeutic effects. This convergence of network pharmacology predictions with experimental validation of oxidative stress modulation, cytokine regulation, and signaling pathway targeting strengthens BRLE’s scientific foundation and supports rational drug development.

BRLE exhibited a comprehensive therapeutic profile through multi-modal mechanisms encompassing clinical improvement, tissue protection, antioxidant activity, and immunomodulation, potentially offering advantages over current single-target IBD therapies. The identification of 200 mg/kg as the optimal dose with minimal toxicity demonstrates clear dose-dependent efficacy and provides valuable guidance for clinical translation. This therapeutic window aligns with the extract’s ability to modulate key inflammatory mediators and restore redox balance, positioning BRLE as a viable adjunct or alternative therapy, particularly when conventional immunosuppressive drugs are poorly tolerated or ineffective.

The antioxidant properties, evidenced by GSH restoration and reduced lipid peroxidation, support mucosal integrity and may help prevent disease flares. Notably, suppression of EGFR, SRC, STAT3, and AKT1, which are implicated in both inflammation and carcinogenesis, opens avenues for investigating BRLE in chemoprevention of colitis-associated colorectal cancer.

Study limitations include the reliance on the acetic acid-induced acute colitis model. While this model effectively simulates the acute mucosal injury and inflammatory infiltration seen in disease flares, it does not capture the chronic relapsing nature of human UC or the complex intestinal flora disorders associated with long-term pathology. Therefore, results regarding long-term maintenance of remission cannot be extrapolated from these data. Additionally, this study focused on mechanistic validation and did not include a positive drug control group (e.g., Sulfasalazine), which limits direct comparisons with standard-of-care therapies. Moreover, while the study validated the modulation of upstream signalling pathways (EGFR, SRC, STAT3), it did not directly quantify downstream structural tight junction proteins (such as ZO-1 or Occludin). Although the reduction in colonic oedema and histological restoration suggests barrier recovery, future studies should specifically evaluate these junctional markers to confirm the direct effect on paracellular permeability. Furthermore, the study utilized male subjects exclusively; given the known influence of sex hormones on inflammatory pathways, these findings may not be fully representative of the female population. Consequently, future investigations should incorporate chronic DSS models, sex as a variable, assessment of systemic inflammatory markers (e.g., plasma cytokines, LPS), comprehensive pharmacokinetic analysis, humanized models, formulation standardization, and early-phase clinical trials to facilitate translational application and explore synergistic effects with existing UC therapies.

## 4. Materials and Methods

### 4.1. Plant Material and Extraction Procedures

Fresh leaves of *Beaucarnea recurvata* (K.Koch & Fintelm.) Lem. were harvested during March 2025 from El-Abd Garden, located in Giza, Egypt. Species identification and taxonomic authentication were performed by Eng. Therese Labib, serving as Plant Taxonomy Consultant at the Ministry of Agriculture and previously holding the position of director at El-Orman Botanical Garden, Giza, Egypt. A reference voucher specimen (ZU-Ph-Cog-0316) was deposited and maintained at the Herbarium of the Pharmacognosy Department, Faculty of Pharmacy, Zagazig University, for future reference.

For extraction purposes, freshly harvested leaves (500 g) were chopped and subjected to maceration using 90% ethanol as the extraction solvent. The maceration process involved three successive extractions (3 × 1 L), each conducted at room temperature (25 ± 2 °C) for 72 h with intermittent stirring at 200 rpm to ensure exhaustive extraction of bioactive constituents. The combined ethanolic extracts were then concentrated under reduced pressure at 40 °C using a rotary evaporator set to −0.08 MPa, yielding 70 g of a viscous extract residue.

### 4.2. Ultra-Performance Liquid Chromatography-Electrospray Tandem Mass Spectrometry (UPLC-ESI-MS/MS) Analysis of BRLE

For analytical characterization, 50 mg of BRLE was solubilized in 1 mL of a ternary solvent system comprising water, methanol, and acetonitrile (50:25:25, *v*/*v*/*v*). The sample underwent mechanical agitation via vortex mixing for 2 min, followed by ultrasonic treatment for 10 min. Centrifugal separation was performed at 1000 rpm for 10 min. A 50 μL aliquot of the supernatant was subsequently diluted with reconstitution buffer to achieve a final volume of 1000 μL. An aliquot of 10 μL, corresponding to 25 μg of extract, was injected for UPLC-ESI-MS/MS analysis in negative ionization mode.

The chromatographic separation was conducted using an ExionLC™ AD UPLC system coupled with a TripleTOF 5600+ Time-of-Flight Tandem Mass Spectrometer (AB SCIEX, AB SCIEX, Framingham, MA, USA; and Concord, ON, Canada, respectively), employing a previously validated methodology [34]. Pre-filtration was achieved using in-line filter discs (0.5 μm × 3.0 mm, Phenomenex^®^, Torrance, CA, USA), while separation was performed on an X-select HSS T3 column (2.5 μm, 2.1 × 150 mm, Waters^®^, Milford, MA, USA) maintained at 40 °C with a flow rate of 0.3 mL/min.

The mobile phase system consisted of two eluents: Eluent A (5 mM ammonium formate buffer, pH 8.0, containing 1% methanol) and Eluent B (100% acetonitrile). A gradient elution profile was implemented: initial conditions maintained 90% Eluent A and 10% Eluent B for 20 min, followed by a transition to 10% Eluent A and 90% Eluent B over 5 min, concluding with a 3 min re-equilibration to starting conditions.

Compound identification was accomplished through comprehensive evaluation of retention time data, molecular mass determination, deprotonated molecular ion ([M-H]^−^) mass-to-charge ratios, and comparative analysis of high-resolution MS and MS/MS spectral data using PeakView™ software version 2.1. Peak integration and quantitative assessment were performed using the Extracted Ion Chromatogram Manager within PeakView software (AB SCIEX, version 1.2.0.3).

### 4.3. Network Pharmacology

#### 4.3.1. Screening of Bioactive Components in BRLE

The canonical SMILES notations for 38 BRLE secondary metabolites detected through LC-MS analysis were obtained from the PubChem repository (https://pubchem.ncbi.nlm.nih.gov/, accessed 20 April 2025) or generated using ChemDraw software v22.0.0.22 (PerkinElmer Informatics, Beaconsfield, UK). These molecular representations were subsequently analyzed through the SwissADME platform (http://www.swissadme.ch/, accessed 21–22 April 2025) [107] to evaluate their pharmacokinetic properties. Compound selection criteria included adherence to Lipinski’s rule of five parameters and demonstration of a bioavailability score ≥ 0.55 [108].

#### 4.3.2. Determination of Common Targets Between UC and BRLE Active Components

Target proteins for the selected bioactive compounds were predicted using the SwissTargetPrediction platform (http://www.swisstargetprediction.ch/, accessed 23 April 2025) [109], with subsequent validation performed through the UniProt repository (https://www.uniprot.org/, accessed 24 April 2025) [110]. Protein nomenclature was standardized according to official gene symbols, and redundant entries were eliminated.

UC-associated targets were retrieved from multiple databases, including GeneCards (https://www.genecards.org/, accessed 25 April 2025) [111], DisGeNeT (https://www.disgenet.org/search, accessed 25 April 2025) [112], and the Online Mendelian Inheritance in Man database (OMIM, https://www.omim.org/, accessed 25 April 2025) [113], using “Ulcerative Colitis” as the search term. Corresponding UniProt identifiers and gene symbols were standardized through UniProt database queries, with duplicate entries removed.

The intersection of compound-associated targets and UC-related genes was determined and visualized using Venny 2.1.0 (https://bioinfogp.cnb.csic.es/tools/venny/, accessed 26 April 2025) to identify common therapeutic targets.

#### 4.3.3. Construction of Protein–Protein Interaction (PPI) Networks

Following target identification, protein interaction networks were established using the STRING database (version 12.0; https://string-db.org/; accessed 27 April 2025) [114]. Analysis parameters included a minimum confidence score of 0.7 and restriction to the Homo sapiens species. The resulting interaction data was imported into Cytoscape software (version 3.10.2; NIGMS, Bethesda, MD, USA) [115] for network visualization and topological analysis.

Network centrality parameters were computed using the CytoNCA plugin, version 2.1, including: betweenness centrality (BC, measuring a node’s role as an intermediary), closeness centrality (CC, assessing node proximity within the network), and degree centrality (DC, quantifying direct node connections) [108,116].

Hub nodes were identified through a hierarchical screening protocol: Initially, nodes displaying DC values ≥ twice the median were selected. Subsequently, core targets were filtered by retaining nodes with BC, CC, and DC values meeting or exceeding their respective median thresholds [117]. This approach emphasized highly connected nodes with central roles in network architecture, potentially representing key regulators in disease mechanisms or therapeutic intervention points.

#### 4.3.4. Development of Compound-Target Interaction Networks

Compound-target interaction networks were constructed using Cytoscape software (v3.10.2) to visualize relationships between BRLE bioactive compounds and their predicted therapeutic targets in UC pathogenesis. Network topology featured compounds and targets as nodes, with edges representing predicted interactions. The CytoHubba plugin [118] was employed for centrality assessment, utilizing degree centrality as the primary ranking parameter. Based on this analysis, the top ten highest-scoring compounds were selected for subsequent investigation due to their potential significance in modulating UC-associated biological pathways.

#### 4.3.5. Functional Annotation Through Gene Ontology and KEGG Pathway Analysis

The 24 hub targets were subjected to comprehensive functional characterization via GO and KEGG pathway enrichment analyses. Target genes were submitted to the DAVID bioinformatics resource (https://davidbioinformatics.nih.gov/; accessed 28 April 2025) [119] for functional annotation across molecular functions (MFs), biological processes (BPs), and cellular components (CCs), alongside KEGG pathway mapping. All analyses were conducted within the Homo sapiens taxonomic framework, employing Benjamini-Hochberg false discovery rate correction for multiple testing (adjusted *p* < 0.05). Results were visualized through bar graphs and bubble plots using the Bioinformatics Online Platform (http://www.bioinformatics.com.cn/en; accessed 28 April 2025) to enhance interpretation and emphasize pathway significance.

### 4.4. Molecular Docking Analysis

To corroborate findings from network analysis, molecular docking investigations were conducted using the four most significant core target proteins and ten principal bioactive compounds from BRLE. Binding affinities were computed to evaluate the strength of molecular interactions. Crystal structures were obtained from the Protein Data Bank (https://www.rcsb.org; accessed 29 April 2025) [120] for the following targets: epidermal growth factor receptor (EGFR; PDB ID: 5U8L; 1.60 Å resolution) [121], proto-oncogene tyrosine-protein kinase Src (SRC; PDB ID: 4MXO; 2.10 Å resolution) [122], signal transducer and activator of transcription 3 (STAT3; PDB ID: 6QHD; 2.85 Å resolution) [123], and RAC-alpha serine/threonine-protein kinase (AKT1; PDB ID: 4EKL; 2.00 Å resolution) [124]. Structural preparation of proteins was performed using UCSF Chimera software (version 1.17.3) [125] according to standard procedures [126].

The 3D conformers of the BRLE phytoconstituents were obtained from PubChem and converted into the pdbqt format using OpenBabel (v2.4.1) [127]. Docking simulations were performed using AutoDock Vina (v1.1.2) integrated into UCSF Chimera, employing default settings with an exhaustiveness of 8 and energy range of 4, generating ten docking poses for each ligand.

For each target protein, the active site was defined using grid boxes centered on the respective binding pockets. Coordinates for the docking grid were obtained using the AutoDock Vina module version 4.2. within UCSF Chimera. Validation of the docking protocol was ensured by re-docking co-crystallized ligands, where available, to confirm the accuracy of active site positioning. In the case of STAT3, which lacks a co-crystallized ligand, the active pocket was predicted using the Computed Atlas for Surface Topography of Proteins (CASTp) server, (http://sts.bioe.uic.edu/castp/index.html?3igg; accessed 30 April 2025) [128], and corresponding grid parameters were defined accordingly. Details of the grid centers, and dimensions are provided in Appendix A.

Binding affinities were expressed as docking scores in kcal/mol, where lower energy values indicate stronger binding propensities. The most favorable docking pose for each compound was selected based on minimal root mean square deviation (RMSD). Molecular interactions, including hydrogen bonding, hydrophobic interactions, and steric complementarity, were visualized and analyzed using Biovia Discovery Studio Visualizer (v21.1.0.20298) [129].

### 4.5. In Vivo Experimental Design

#### 4.5.1. Experimental Animals, Ethical Compliance, and Study Design

The experimental protocol received institutional approval from the Zagazig University Institutional Animal Care and Use Committee (approval number: ZU-IACUC/3/F/17/2025). All procedures adhered to the Animal Research: Reporting of In Vivo Experiments (ARRIVE) guidelines and institutional animal welfare standards.

Adult male Wistar rats weighing 180–220 g were procured from the animal facility at the Faculty of Veterinary Medicine, Zagazig University. Prior to experimentation, animals underwent a seven-day acclimatization period under controlled environmental conditions: 12:12 h light-dark photoperiod, ambient temperature of 22 ± 2 °C, with ad libitum access to standard laboratory chow and water.

Following randomization, twenty-five rats were assigned to five experimental cohorts (*n* = 5 per group). The study included two control groups and three treatment groups. Group 1 served as the normal control, receiving an intrarectal administration of 2 mL of physiological saline on day zero. Group 2, designated as the disease control, underwent ulcerative colitis induction via a single intrarectal instillation of 2 mL acetic acid solution (3% *v*/*v* in normal saline) on day zero, followed by vehicle administration.

The treatment groups (Groups 3–5) were subjected to ulcerative colitis induction using the same acetic acid protocol on day zero, followed by oral administration of *Beaucarnea recurvata* leaf extract (BRLE) at escalating doses of 100, 200, and 400 mg/kg/day, respectively. BRLE treatment commenced 2 h after UC induction and was continued once daily for six consecutive days via oral gavage

Vehicle preparation involved dissolving BRLE in 1% Tween 80 solution (*v*/*v* in distilled water), prepared fresh daily. Control groups received equivalent volumes of vehicle solution to maintain experimental consistency. The acetic acid solution (3% *v*/*v*) was prepared aseptically in sterile normal saline (0.9% NaCl).

Animals were subjected to 24 h fasting prior to colitis induction. Under ketamine anesthesia (50 mg/kg, i.p.), a sterile 6F polypropylene catheter was carefully inserted 6–8 cm into the rectum with animals positioned in the Trendelenburg position. Following instillation, animals were maintained inverted for 30 s to prevent solution reflux, as adapted from established protocols [31].

#### 4.5.2. Animal Welfare and Monitoring

Daily clinical assessments included behavioral observation, body weight monitoring, and stool examination. Predetermined endpoints for early termination encompassed severe weight loss (>20%), persistent hemorrhage, or systemic complications requiring immediate intervention.

#### 4.5.3. Sample Collection and Preparation

Twenty-four hours following the final treatment dose, animals were euthanized under deep ketamine anesthesia (50 mg/kg, i.p.) with confirmation of death verified through cardiac examination and exsanguination. Colonic tissues were rapidly excised, irrigated with ice-cold phosphate-buffered saline (pH 7.4), and processed accordingly. Distal colon segments were preserved in 10% neutral buffered formalin for histopathological evaluation, while the remaining tissue was homogenized in ice-cold PBS (1:9 weight/volume ratio), centrifuged at 12,000× *g* for 15 min at 4 °C, with supernatants collected and stored at −80 °C for subsequent biochemical analyses.

#### 4.5.4. Disease Severity Assessment

Disease progression was monitored using a composite Disease Activity Index (DAI) scoring system [130], evaluating three key parameters as revealed in Table 3.

The composite DAI was calculated as the arithmetic mean of the three component scores:DAI = (Weight Loss Score + Stool Score + Bleeding Score) ÷ 3

#### 4.5.5. Macroscopic Pathological Evaluation

Post-mortem colonic examination included visual assessment of inflammatory changes using a standardized macroscopic scoring system [131], as displayed in Appendix A.

For assessment of tissue inflammation, colons were carefully weighed and their length precisely measured using a standard ruler under consistent tension to avoid stretching artifacts. The colon weight/length ratio was subsequently calculated as an additional inflammatory biomarker.

#### 4.5.6. Biochemical and Molecular Marker Analysis

Tissue homogenates were analyzed for oxidative stress indicators, inflammatory mediators, and network pharmacology targets:

##### Oxidative Stress Parameters

Oxidative stress parameters were evaluated by quantifying reduced glutathione (GSH) and malondialdehyde (MDA) levels in tissue homogenates. GSH was measured using an enzymatic cycling colorimetric assay (BioVision K464-100, Biovision Inc., Milpitas, CA, USA), with absorbance read at 450 nm. Lipid peroxidation was assessed by determining MDA levels via the thiobarbituric acid reactive substances (TBARS) assay (BioVision K739-100, Biovision Inc., Milpitas, CA, USA), with absorbance measured at 532 nm.

##### Inflammatory Cytokines

Inflammatory cytokines were quantified to assess the immune response profile. Tumor necrosis factor-alpha (TNF-*α*) levels were measured using a sandwich ELISA method (BT-Laboratory E0764Ra, BT-Laboratory, Shanghai, China), with a detection limit of 2.51 ng/L. Interleukin-10 (IL-10) concentrations were determined by ELISA (BT-Laboratory E0108Ra, BT-Laboratory, Shanghai, China), exhibiting a sensitivity of 1.51 pg/mL.

##### Network Pharmacology Targets

To investigate molecular targets implicated in the proposed pharmacological mechanisms, a panel of key signaling proteins was quantified using ELISA-based methods. Epidermal growth factor receptor (EGFR) levels were measured using a sandwich ELISA kit (Elabscience E-EL-R0369, Houston, TX, USA), while SRC protein concentrations were determined via an ELISA assay (Abbexa abx547297, Cambridge, UK). Signal transducer and activator of transcription 3 (STAT3) was quantified using a rat-specific sandwich ELISA (MyBioSource MBS760293, San Diego, CA, USA), with a detection range of 0.156–10 ng/mL and a sensitivity of <0.094 ng/mL. Additionally, AKT1 kinase levels were assessed using a colorimetric ELISA kit (Novus Biologicals NBP3-42380, Centennial, CO, USA).

All assays were performed in triplicate with concentrations determined from respective standard curves.

#### 4.5.7. Histopathological Analysis

Formalin-preserved tissues underwent standard histological processing: dehydration through graded ethanol series, xylene clearing, and paraffin embedding. Serial sections (5 μm thickness) were prepared using automated microtomy and stained with hematoxylin and eosin for light microscopic examination [132]. To provide a quantitative assessment of mucosal injury, the tissue sections were evaluated according to the Geboes Score (Table 4), a validated histological grading system for ulcerative colitis [133]. The system evaluates five histological features: structural change, chronic inflammatory infiltrate, lamina propria eosinophils, lamina propria neutrophils, and neutrophils in the epithelium. Each parameter is graded on a scale (0–3 or 0–4), with higher scores indicating severe inflammation and tissue damage.

#### 4.5.8. Statistical Methodology

Data are presented as mean ± standard error of the mean (SEM). Statistical comparisons utilized one-way analysis of variance (ANOVA) with Tukey’s multiple comparison post hoc analysis. All statistical computations and graphical representations were generated using GraphPad Prism software (Version X, GraphPad Software Inc., San Diego, CA, USA). Statistical significance was established at *p* < 0.05.

## 5. Conclusions

This study highlights the therapeutic potential of *Beaucarnea recurvata* leaf extract (BRLE) in the management of ulcerative colitis. UPLC-ESI-MS/MS analysis revealed a rich phytochemical composition, including steroidal saponins, triterpenes, flavonoids, and phenolic acids, which collectively contribute to its multi-target biological activity. BRLE demonstrated significant anti-inflammatory, antioxidant, and tissue-protective effects, primarily through modulation of key inflammatory signaling proteins such as EGFR, SRC, STAT3, and AKT1. The 200 mg/kg dose showed the most consistent efficacy, improving clinical scores, restoring redox balance, normalizing cytokine levels, and enhancing mucosal healing. Notably, BRLE downregulated several molecular pathways involved in both inflammation and tumor progression. Overall, BRLE represents a promising adjunct or alternative therapy for UC, particularly in cases with poor response or intolerance to standard treatments. Further studies in chronic models, pharmacokinetic assessments, and clinical trials are needed to support its translation into therapeutic use.

## Figures and Tables

**Figure 1 ijms-26-12053-f001:**
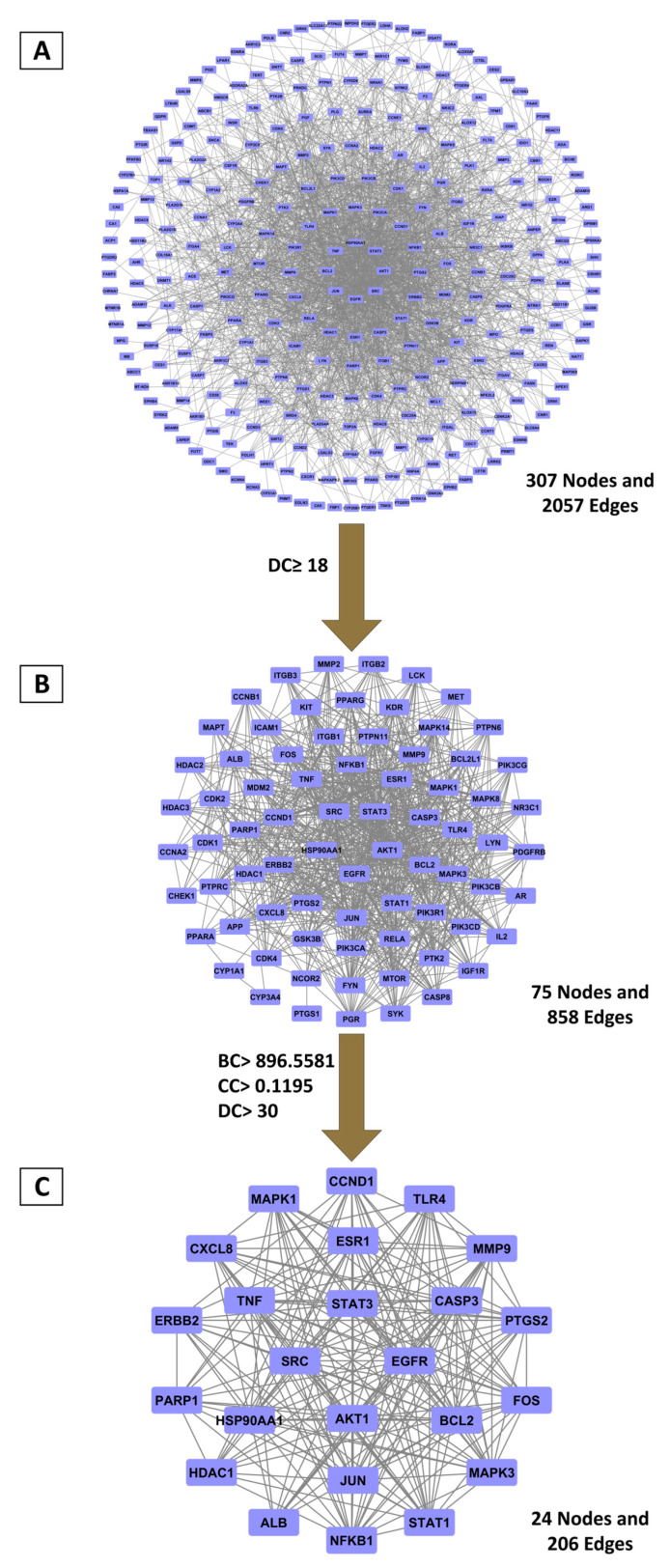
Principal targets derived from PPI network topology analysis. (**A**) Complete PPI network encompassing 307 overlapping targets; (**B**) Primary selection applying double-median degree centrality threshold; (**C**) Core targets selected based on betweenness, closeness, and degree centrality measures exceeding median values.

**Figure 2 ijms-26-12053-f002:**
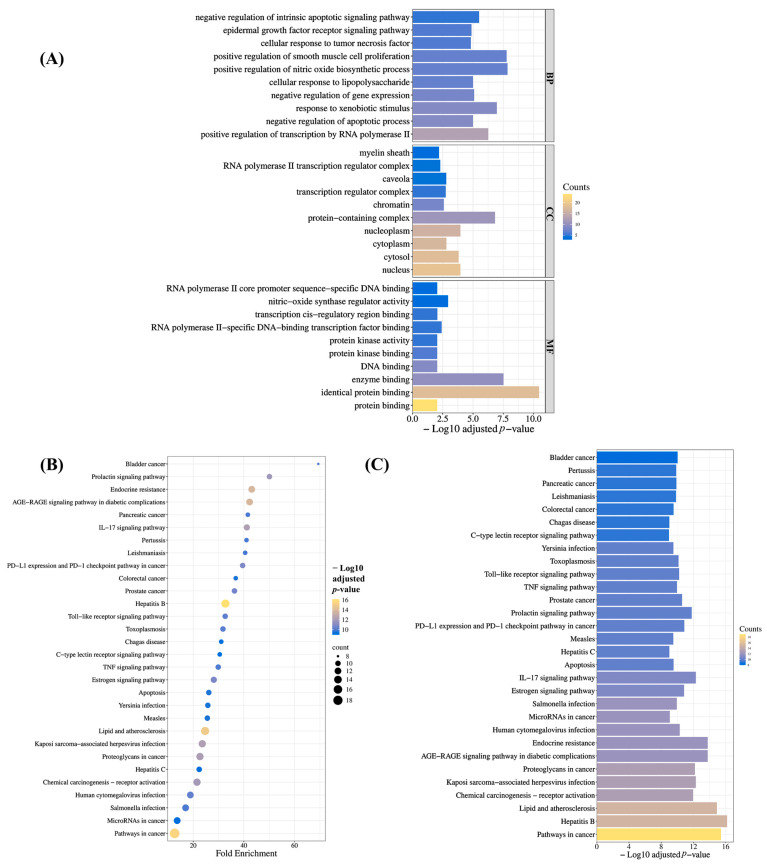
Functional enrichment analysis outcomes. (**A**) Significantly enriched Gene Ontology (GO) categories across biological processes (BPs), cellular components (CCs), and molecular functions (MFs). (**B**) Bubble plot and (**C**) bar chart representation of enriched Kyoto Encyclopedia of Genes and Genomes (KEGG) pathways.

**Figure 3 ijms-26-12053-f003:**
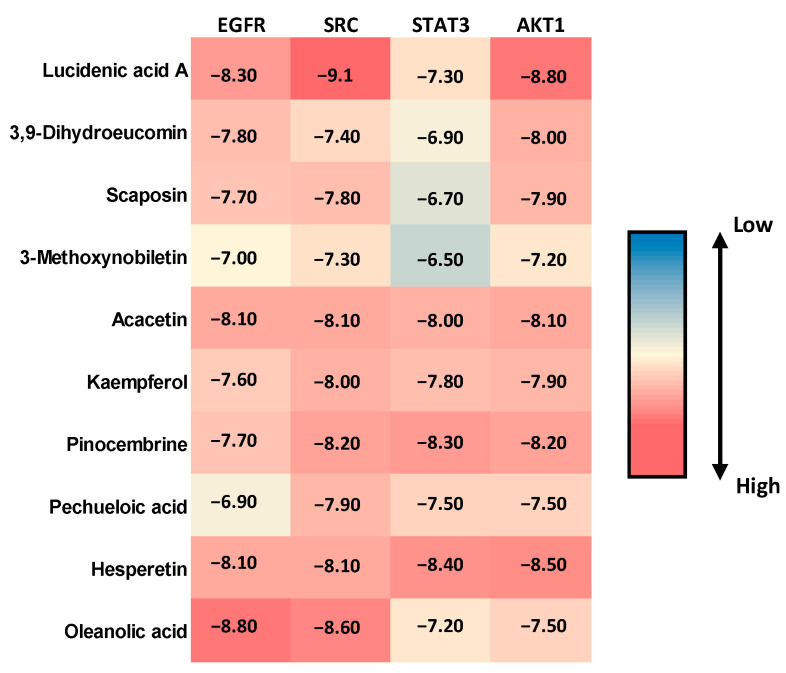
Heatmap visualization of molecular docking scores (kcal/mol) for the top ten bioactive compounds from *Beaucarnea recurvata* leaf extract (BRLE) against four ulcerative colitis-related protein targets (EGFR, SRC, STAT3, and AKT1). Darker shades represent stronger (more negative) binding affinities, indicating potential for multi-target interaction.

**Figure 4 ijms-26-12053-f004:**
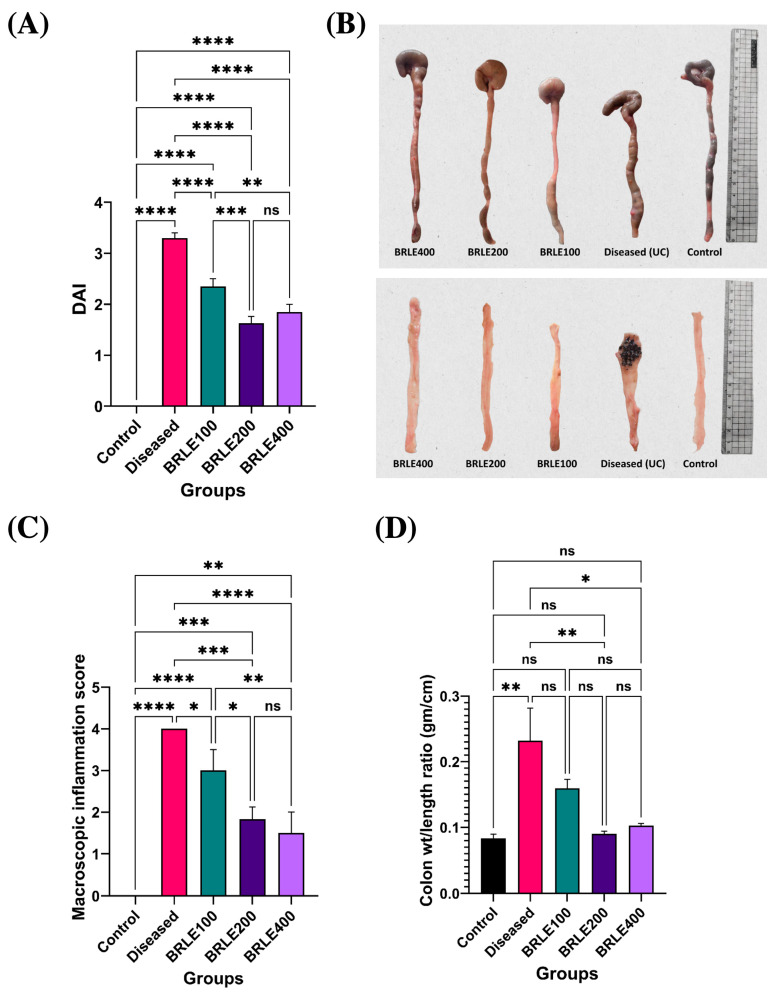
Effect of *Beaucarnea recurvata* leaf extract (BRLE) at doses of 100, 200, and 400 mg/kg/day on Disease Activity Index (DAI) and inflammatory parameters in acetic acid-induced ulcerative colitis in rats. (**A**) Representative macroscopic images of isolated rat colons from each group illustrating differences in colonic length, thickness, and edema. (**B**) Visual comparison of dissected and cleaned colonic segments post-longitudinal opening, demonstrating mucosal integrity and severity of surface inflammation or ulceration across experimental groups. (**C**) Macroscopic inflammation score assessment. (**D**) Colon weight-to-length ratio analysis. Data are presented as mean ± SD (*n* = 5/group). Statistical comparisons were performed using one-way ANOVA followed by Tukey’s post hoc test. **** *p* < 0.0001, *** *p* < 0.001, ** *p* < 0.01, * *p* < 0.05, ns = non-significant.

**Figure 5 ijms-26-12053-f005:**
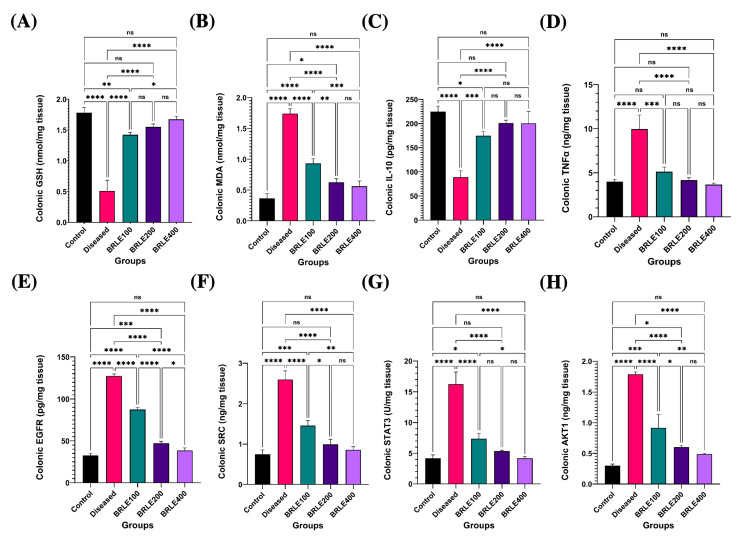
Effect of *Beaucarnea recurvata* leaf extract (BRLE) at doses of 100, 200, and 400 mg/kg/day on colonic oxidative stress markers, cytokine levels, network pharmacology targets and histopathological features in rats with acetic acid-induced ulcerative colitis. (**A**) GSH: Colonic content of reduced glutathione. (**B**) MDA: Colonic malondialdehyde levels. (**C**) IL-10: Levels of the anti-inflammatory cytokine interleukin-10. (**D**) TNF-*α*: Levels of the pro-inflammatory cytokine tumor necrosis factor-alpha. (**E**) EGFR: Expression level of epidermal growth factor receptor. (**F**) SRC: Expression level of SRC kinase. (**G**) p-STAT3: Expression level of phosphorylated STAT3. (**H**) AKT1: Expression level of AKT1 protein. Data are expressed as mean ± SEM (*n* = 5/group). Statistical analysis was performed using one-way ANOVA followed by Tukey’s post hoc test. **** *p* < 0.0001, *** *p* < 0.001, ** *p* < 0.01, * *p* < 0.05; ns = not significant. (**I**–**M**) Representative photomicrographs of H&E-stained colonic sections; (**I**) control group showing normal histological architecture. Key features include the lamina epithelialis (arrowhead), lamina propria and muscularis mucosa (arrow), submucosa (red star), muscular layer (black star), and serosa. (**J**) diseased group showing ulcerative colitis with remnant of necrotic surface epithelium (red arrowhead) adjacent to cystically dilated some colonic crypts (red curved arrow). Heavy infiltration of mucosa, submucosa by eosinophils (red thick arrow), macrophages (yellow thick arrow), lymphocytes (yellow thin arrow), fibrin threads, and few neutrophils beside edema (red star) and dilated submucosal vasculature (red thin arrow). Necrotic myocytes with pyknotic nuclei and infiltrated with abundant numbers of inflammatory cells at muscular layer. (**K**) BRLE100-treated group showing mucous exudate adhered to colonic epithelium (yellow arrowhead), still present ulcerated mucosal surface with appearance of tunica muscularis toward luminal surface (red arrowhead). Hyalinized or necrotic tunica muscularis beside fibrin threads & few inflammatory cells at submucosal layer (yellow star). Exudation at serosal layer. (**L**) BRLE200-treated group showing regenerated colonic surface epithelium (arrowhead), subepithelial granulation tissue infiltrated with moderate numbers of inflammatory cells (yellow star), hyperplastic or regenerated colonic crypts (red thin arrow) beside some cystically dilated colonic crypts (red curved arrow), edema with few inflammatory cells infiltrates submucosal layer (black star). (**M**) BRLE400-treated group showing healed ulcerated area with desquamated mucosal epithelium (red arrowhead), inflammatory cells infiltrates in between colonic crypts, crypt dilatation (red curved arrow), submucosal edema (yellow star), mild number of inflammatory cells within muscular layer beside serosal exudates. Scale bar: 100 μm.

**Table 1 ijms-26-12053-t001:** Comprehensive phytochemical characterization of *Beaucarnea recurvata* leaf ethanolic extract via UPLC-ESI-MS/MS in negative ionization mode.

No.	Rt.	[M-H]^−^	MS^2^ Fragments (*m*/*z*)	Tentative Identification	Class	Reference
1.	1.262	181.073	163, 145, 127, 101, 89	Hexitol (D-sorbitol)	Sugar alcohol	[26]
2.	1.274	151.062	133, 131, 119, 89, 59	Pentitol (xylitol)	Sugar alcohol	[27]
3.	1.361	135.031	117, 91, 73	Threonic acid	Sugar acid	[28]
4.	1.398	149.047	131, 59	D-arabinose	Monosaccharide	[29]
5.	1.493	267.075	249, 135, 108, 92	Inosine	Nucleoside	[30]
6.	1.515	133.015	115, 89, 71, 59	Malic acid	Organic acid	[31]
7.	1.529	191.021	147, 173	D-glucaro-1,4-lactone	Sugar acids derivative	[32]
8.	1.571	179.020	161, 143, 125, 107, 89, 71	Glucose	Monosaccharide	[15]
9.	2.192	325.117	193, 149	Ferulic acid pentoside (feruloyl-arabinose)	Phenolic acid dv.	[33]
10.	2.537	421.002	191, 111	Malonylcoumaroylquinic acid	Phenolic acid dv.	[10]
11.	2.882	117.020	99, 73	Succinic acid	Omega-dicarboxylic acid	[34]
12.	3.946	161.047	143, 125, 101, 117, 99, 73	3-Hydroxy-3-methylglutaric acid [Meglutol]	Dicarboxylic acid	[35]
13.	4.555	164.073	147, 119, 103	D-phenylalanine	Amino acid	[36]
14.	4.704	487.172	163, 145	Coumaric acid dihexoside	Phenolic acid dv.	[10]
15.	5.828	153.021	109	2,3-Dihydroxybenzoic acid	Phenolic acid	[36]
16.	5.921	195.031	177, 151	Gluconic acid	Gluconic acid	[10]
17.	5.967	529.126	349, 193	Caffeoyl–feruloylquinic acid	Phenolic acid dv.	[14]
18.	6.125	129.020	111, 85	Monomethyl fumarate	Dicarboxylic acid	[37]
19.	6.444	151.041	133, 121, 107	4-hydroxyphenylacetic acid	Monocarboxylic acid	[38]
20.	6.674	175.063	157, 131, 129, 113, 115, 87, 85	2-Isopropylmalic acid	Dicarboxylic acid	[39]
21.	6.680	457.178	411, 163	Lucidenic acid A	Triterpenoid	[14]
22.	6.783	145.051	127, 101	Adipic acid	Dicarboxylic acid	[40]
23.	6.879	179.083	161, 135	Caffeic acid	Phenolic acid	[34]
24.	6.921	137.025	119, 108	3,4-Dihydroxybenzaldehyde (Protocatechuic aldehyde)	Phenolic aldehyde	[41]
25.	7.007	201.115	183, 157, 139, 113	Sebacic acid	Alpha,omega-dicarboxylic acid	[42]
26.	7.097	137.026	93	Salicylic acid	Phenolic acid	[43]
27.	7.250	187.099	169, 143, 125, 115	Azelaic acid	Alpha,omega-dicarboxylic acid	[44]
28.	7.309	593.158	503, 473	Apigenin 6,8-di-C-glucoside	Flavonoids	[10]
29.	7.525	178.053	134	4-Acetamidobenzoic acid (Acedoben)	4-Aminobenzoic acid dv.	[45]
30.	7.692	563.146	503, 473, 443	Apigenin 6-C-glucoside 8-C-arabinoside	Flavonoids	[46]
31.	8.048	579.262	519, 459	Luteolin 6-C-β-glucopyranoside-8-C-α-arabinopyranoside (Carlinoside)	Flavonoids	[10]
32.	8.130	159.067	141, 115, 71	3,3-Dimethylglutaric acid	Alpha,omega-dicarboxylic acid	[47]
33.	8.133	345.099	301, 167, 139	Aucubin	Terpenoid	[14]
34.	8.497	577.164	487, 457	Apigenin 6-C-β-glucopyranoside-8-C-α-rhamnopyranoside (Violanthin) [Vitexin 2″-O-rhamnoside]	Flavonoids	[10]
35.	8.548	299.117	178, 150, 122	Dihydroeucomin	Flavonoids	[15]
36.	8.671	283.124	265, 239	Acacetin	Flavonoids	[14]
37.	8.786	163.041	119	P-coumaric acid	Phenolic acid	[43]
38.	8.808	173.083	155, 129, 111, 85	Suberic acid (Octanedioic acid)	Alpha,omega-dicarboxylic acid	[48]
39.	9.021	167.036	152, 108	Methyl protocatechuate [Protocatechuic acid methyl ester]	Phenolic acid dv.	[49]
40.	9.280	161.025	133	Umbelliferone	Coumarin	[50]
41.	9.575	195.031	180, 165, 151	Homoveratric acid (3,4-Dimethoxyphenylacetic acid)	Phenylacetic acid	[51]
42.	9.924	147.046	129, 117	2-Hydroxyl cinnamaldehyde	Cinnamaldehyde	[52]
43.	10.313	207.068	177, 192, 121	Sinapaldehyde	Cinnamaldehyde	[52]
44.	10.589	389.166	371, 361, 353, 343	Scaposin	Flavone	[14]
45.	10.874	903.345	433[Furostane-triol]	Furostane-triol; rhamnosyldihexoside	Steroidal saponins	[8]
46.	11.673	301.076	286, 164, 151	Hesperetin	Flavonoids	[53]
47.	11.828	431.142	387	3-Methoxynobiletin	Flavonoids	[54]
48.	11.914	255.126	211, 151	Pinocembrine	Flavonoids	[55]
49.	12.013	503.271	179, 135	Caffeic acid dihexoside	Phenolic acid dv.	[10]
50.	12.235	169.087	125	Gallic acid	Phenolic acid	[15]
51.	12.894	769.415	329[aglycone-H]	Dihydroxypregna-5,16-dien-20-one deoxyhexoside pentoside hexosid	Pregnane steroidal saponin	[14]
52.	13.052	247.137	203	Pechueloic acid	Terpenoid	[14]
53.	13.304	421.065	341, 135	Caffeic acid hexosidedv.	Phenolic acid dv.	[10]
54.	13.661	285.211	267, 257, 241	Kaempferol	Flavonoids	[36]
55.	14.922	837.439	705, 559	Neoruscogenin deoxyhexoside dipentoside isomer	Spirostane steroidal saponin	[14]
56.	15.465	837.454	691, 559	Neoruscogenin deoxyhexoside dipentoside	Spirostane steroidal saponin	[14]
57.	15.893	503.273	153, 109	Protocatechuic acid dv.	Phenolic acid dv.	[43]
58.	15.915	455.244	437, 411	Oleanolic acid	Pentacyclic triterpenoid	[56]
59.	16.474	455.217	437, 411	Ursolic acid	Pentacyclic triterpenoid	[56]
60.	16.737	901.508	323	Spirostan-3-ol; Glucopyranosyl-glucopyranosyl galactopyranoside	Spirostane steroidal saponin	[15]
61.	16.919	315.128	297, 282, 78	Eucomol	Flavonoids	[15]
62.	18.144	429.303	383, 344, 345	Ruscogenin	Spirostane steroidal saponin	[14]
63.	19.033	311.172	183, 119	Caftaric acid	Cinnamic acid derivative	[14]
64.	20.205	609.281	447	Isoorientin-7-O-β-glucopyranoside	Flavonoids	[10]
65.	24.466	489.364	447, 357	O-acetylisoorientin	Flavonoids	[10]
66.	28.088	919.275	433[Furostane-triol]	Furostane-triol; trihexoside	Steroidal saponins	[15]

**Table 2 ijms-26-12053-t002:** Scoring system for evaluation common observed lesions in colon among different groups according to Geboes histological score for ulcerative colitis.

Lesion	Control	Disease	BRLE100	BRLE200	BRLE400
Structural change	0, 0, 0, 0, 0	1, 3, 2, 3, 3	3, 2, 2, 1, 1	2, 2, 1, 1, 1	2, 1, 0, 0, 0
Chronic inflammatory infiltrate	0, 0, 0, 0, 0	2, 3, 2, 3, 2	2, 2, 2, 1, 1	2, 1, 1, 1, 0	1, 1, 0, 0, 0
Lamina propria neutrophils	0, 0, 0, 0, 0	1, 3, 1, 2, 2	2, 3, 1, 1, 1	2, 2, 1, 0, 0	2, 1, 0, 0, 0
Lamina propria eosinophils	0, 0, 0, 0, 0	1, 3, 2, 2, 2	2, 2, 1, 1, 1	2, 2, 1, 1, 0	1, 1, 0, 0, 0
Neutrophils in epithelium	0, 0, 0, 0, 0	1, 3, 3, 1, 2	2, 2, 2, 1, 0	1, 1, 1, 1, 0	1, 1, 1, 0, 0
Crypt destruction	0, 0, 0, 0, 0	1, 3, 2, 3, 2	3, 2, 2, 1, 0	2, 2, 1, 1, 0	2, 1, 1, 0, 0
Grade 5 Erosion or ulceration	0, 0, 0, 0, 0	2, 2, 3, 4, 4	4, 3, 2, 2, 1	3, 2, 1, 1, 0	3, 1, 1, 0, 0

**Table 3 ijms-26-12053-t003:** Disease Activity Index (DAI) scoring system used to assess ulcerative colitis severity in rats, based on weight loss, stool consistency, and rectal bleeding. The final DAI score is calculated as the arithmetic mean of the three individual parameters.

Parameter	Score 0	Score 1	Score 2	Score 3	Score 4
Weight Loss (%)	0	1–5	6–10	11–20	>20
Stool Consistency	Normal	—	Loose	—	Liquid
Rectal Bleeding	Absent	Minimal	Evident	—	—

**Table 4 ijms-26-12053-t004:** Different grades used for evaluation severity of ulcerative colitis.

**Grade 0: Structural change**	**Subgrades**
	0.0 No abnormality
	0.1 Mild alteration
	0.2 Mild or moderate diffuse or multifocal alteration
	0.3 Severe diffuse or multifocal alteration
**Grade1: Chronic inflammatory infiltrate**	**Subgrades**
	1.0 No increase
	1.1 Mild but unequivocal increase
	1.2 Moderate increase
	1.3 Marked increase
**Grade 2: Lamina propria neutrophils and eosinophils**	**2A Eosinophils 2A.**
	0 No increase 2A.
	1 Mild but unequivocal increase 2A.
	2 Moderate increase
	3 Marked increase
	**2B Neutrophils 2B.**
	0 None 2B.
	1 Mild but unequivocal increase 2B.
	2 Moderate increase 2B.
	3 Marked increase
**Grade 3**: **Neutrophils in epithelium**	3.0 None
	3.1 <5% crypts involved
	3.2 <50% crypts involved
	3.3 >50% crypts involved
**Grade 4**: **Crypt destruction**	4.0 None
	4.1 Probable—local excess of neutrophils in part of crypt
	4.2 Probable—marked attenuation
	4.3 Unequivocal crypt destruction
**Grade 5**: **Erosion or ulceration**	5.0 No erosion, ulceration, or granulation tissue
	5.1 Recovering epithelium+adjacent inflammation
	5.2 Probable erosion—focally stripped
	5.3 Unequivocal erosion
	5.4 Ulcer or granulation tissue

## Data Availability

The original contributions presented in this study are included in the article and Appendix A. Further inquiries can be directed to the corresponding author.

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
