# Peer review of "Therapeutic Potential of *Beaucarnea recurvata* Leaf Extract Against Ulcerative Colitis: Integrating Phytochemical Profiling, Network Pharmacology, and Experimental Validation"

_ijms, 2025, doi:10.3390/ijms262412053_

Round 1
Reviewer 1 Report
Comments and Suggestions for Authors
This study focuses on the therapeutic potential of Beaucarnea recurvata Leaf Extract (BRLE) for ulcerative colitis (UC). It integrates phytochemical analysis, network pharmacology, molecular docking, and in vivo experimental validation, providing multi-dimensional evidence for the intervention of UC with natural products. However, there is still room for optimization in the paper regarding experimental design details, methodological rigor, in-depth result analysis, and discussion extension. Specific issues are as follows:
- Only male Wistar rats were used in the experiment (Section 4.5.1), with no female animals included. It is known that there are gender differences in the pathogenesis and drug response of UC. It is recommended to clearly state in the discussion that "results from male animals alone may not be representative of the female population" and emphasize that subsequent studies should incorporate gender as a variable.
- An acetic acid-induced acute colitis model was adopted (Section 4.5.1). However, clinical UC is characterized by chronic relapsing inflammation, and the acute model cannot simulate core pathological processes of UC such as repeated intestinal mucosal injury-repair and intestinal flora disorder. It is recommended to supplement experiments using a dextran sulfate sodium (DSS)-induced chronic UC model.
- The qualitative accuracy of phytochemical analysis is insufficient. It is recommended to purchase reference standards for core active components (e.g., Lucidenic acid A, oleanolic acid, hesperetin) and confirm their structures via UPLC-ESI-MS/MS through the comparison of retention time (Rt) and MS² spectra between reference standards and samples.
- The expression levels of key targets (e.g., p-STAT3) were only detected by ELISA. It is recommended to supplement Western blot analysis to verify the expression of EGFR, SRC, p-STAT3, and AKT1.
- It is recommended to quantify H&E-stained sections using recognized quantitative standards (e.g., Geboes scoring system, Matts scoring system) and supplement the scoring data of each group.
- In the discussion section, it is recommended to supplement the analysis of key differences between BRLE and extracts from congeneric plants to highlight the research innovation.
- Please cite the following references, “In Silico Studies as Support for Natural Products Research. Medinformatics (Google Scholar), ADME, Molecular Targets, Docking and Dynamic Simulation Studies of Phytoconstituents of Cymbopogon citratus (DC.). Medinformatics (Google Scholar)”.
Reviewer 2 Report
Comments and Suggestions for Authors
This manuscript addresses a research topic of certain value, and its integrated research approach is relatively innovative.
However, there are several aspects that require revisions, and substantial modifications are needed.
- The manuscript mentions that 500 g of fresh leaves were subjected to 3 successive extractions using 90% ethanol, yielding 70 g of a viscous extract. Nevertheless, critical parameters such as extraction temperature and extraction time were not specified. Different extraction conditions can significantly affect the yield and composition of bioactive components. It is recommended to supplement the complete extraction process parameters, including the duration, temperature, and stirring speed for each extraction, as well as the temperature and pressure conditions for vacuum concentration.
- The current experiment only includes a normal control group, a disease control group, and three BRLE dose groups, with the absence of a positive drug control group.
- A total of 66 secondary metabolites were identified via UPLC-ESI-MS/MS, but Table 1 only presents information on some of these compounds. Additionally, the relative content or peak area percentage of each compound was not specified, making it impossible to determine the contribution of major bioactive components.
- In the conclusions, it is stated that "BRLE may also reduce the risk of developing colitis-associated colorectal cancer." However, the study did not involve the detection of any cancer-related indicators, and this conclusion lacks experimental evidence. It is recommended to either remove this part of the conclusion or supplement relevant experimental data to support this speculation.
- The study used an acute acetic acid-induced model to evaluate the therapeutic potential of BRLE for UC. However, this model exhibits significant differences from the pathophysiological characteristics of human UC, which may affect the clinical translational value of the study conclusions.
- The core pathological features of the acetic acid-induced model include acute oxidative stress and mucosal necrosis. Yet, the study only detected oxidative stress indicators such as GSH and MDA, which is insufficient to support the mechanistic conclusion that "BRLE regulates UC-related pathways." It is recommended to supplement the detection of mRNA or protein expression of relevant pathway molecules to improve the validation of model effectiveness.
Reviewer 3 Report
Comments and Suggestions for Authors
Tawfeek et al. present a well organized paper to explore the utility of the Beaucarnea recurvata extract against Ulcerative colitis by analyzing the components of the extract, a well thought bioinformatics network analysis as well as molecular docking to find the mechanistic relevance of these extracts on UC.
While most aspects are convincing, the Ulcerative colitis model results are overstated atleast in the main manuscript. It needs more supporting data, and the histopathology images, though very detailed in explanation is lacking in clarity of the images.
- The arrows point out to important things in theory- but when zoomed in, it is just pixelated. It will be useful to take a higher magnification image of all the areas that the authors like to feature: basically all the ones pointed with arrows. Since so much emphasis is placed on the histology, please bring the supplementary data to the main figures.
- The pointing out to epithelial erosions, and other broken crypts could easily be an artefact of several issues: fixation delays, slicing the paraffin blocks, etc. So these should not be overstated.
- It looks like there is a dose curve and not a straight dose dependency according to the H&Es but not according to the assays. In other words, 400mg/kg dose is most beneficial as demonstrated by the cytokine assays, but histopathologically, 200mg/kg does better.
- Tight junctions and adherens junctions are the bearers of paracellular permeability of the intestinal epithelium. It will be prudent to see Occludin and/or ZO-1 staining compared between the different groups. Especially, in ulcerative colitis, one is bound to see ulcerated patches amongst non-ulcerated, normal looking tissue. But there may still be weakening of the epithelial paracellular junctions.
- Was any plasma analyzed for cytokines and LPS? this will provide a better proof of the extracts working.
- The authors can include a discussion on the importance of the individual (top 10) bioactive compounds on how they have been (or not been) used therapeutically, and perhaps a comparison of the differences between the bioactive compound distribution variability amongst B. recurvata grown elsewhere in different soil and climatic conditions would be enlightening.
Round 2
Reviewer 2 Report
Comments and Suggestions for Authors
The author has solved these problems.